# Lorentz Symmetry and High-Energy Neutrino Astronomy

Carlos A. Argüelles [1] and Teppei Katori [2,*]

1 Department of Physics & Laboratory for Particle Physics and Cosmology, Harvard University, Cambridge, MA 02138, USA; carguelles@fas.harvard.edu
2 Department of Physics, King's College London, London WC2R 2LS, UK
* Correspondence: teppei.katori@kcl.ac.uk

**Abstract:** The search of the violation of Lorentz symmetry, or Lorentz violation (LV), is an active research field. The effects of LV are expected to be very small, and special systems are often used to search it. High-energy astrophysical neutrinos offer a unique system to search signatures of LV, due to the three factors: high neutrino energy, long propagation distance, and the presence of quantum mechanical interference. In this brief review, we introduce tests of LV and summarize existing searches of LV, using atmospheric and astrophysical neutrinos.

**Keywords:** Lorentz and CPT symmetry; standard-model extension; quantum gravity; astrophysical neutrinos; IceCube





## 1. Tests of Lorentz Violation with High-Energy Astrophysical Neutrinos

Lorentz symmetry is a fundamental symmetry, underlying both quantum field theory and general relativity. Nevertheless, the violation of Lorentz symmetry, often known as Lorentz violation (LV), has been searched for since the iconic Michelson–Morley experiment [1]. LV has shown to occur in beyond the standard model (BSM) theories, such as string theory [2], non-commutative field theory [3], loop quantum gravity [4], Hořava–Lifshitz gravity [5], etc. There are many experimental efforts to search for LV, but, so far, no significant evidence for LV has been found. Constraints obtained from different systems have being compiled in Ref. [6]. Since the expected effect of LV is small, experiments tend to use special systems to maximize their sensitivities, such as interferometers (optics [7], matter wave [8], wave function [9], etc.). High-energy particles (LHC [10], high-energy gamma rays [11], ultra-high-energy cosmic rays, or UHECRs [12], etc.) are used to search for signatures of high-dimension LV operators [13–15] that have mass dimensions greater than four.

Among the many experiments hunting for LV, the searches using high-energy astrophysical neutrinos are special for the following three reasons:

1. Neutrino energy reaches higher than any anthropogenic beam.
2. Neutrinos travel very long distance, from source to detection, in a straight path.
3. Quantum mixings can enhance the sensitivity.

LV can be seen as a classical æther field, a new background field permeating the vacuum. Propagation of neutrinos may be affected by this which can cause a variety of effects, including spectrum distortion, modifying the group velocity, anomalous neutrino oscillations, with the direction dependence. Astrophysical neutrinos propagate long distances without interactions, and they have the advantage to search for these exotic effects. Furthermore, the higher-dimension operators, the nonrenormalizable sector of effective field theory, have stronger energy dependence, and high-energy astrophysical neutrinos can be more sensitive to them. For example, the dimension-six operator is the lowest-order interaction term with new physics. Lastly, these effects are likely to be very small, and kinematic tests may not be sensitive enough to find them. Neutrinos are natural

interferometers, and, using quantum mixings of them, we can reach the signal region of LV expected from quantum-gravity motivated models.

Figure 1 shows the phase space of the new physics one can explore with neutrinos [16]. Here, the horizontal axis is the neutrino energy, and the vertical axis is the propagation distance of neutrinos from source to detector. Large areas, below 100 GeV and 100 km, are explored by anthropogenically produced neutrinos (e.g., reactor, short- and long-baseline, or SBL and LBL neutrino experiments) and low-energy astrophysical neutrinos (solar and supernova neutrinos). However, higher energy and longer baseline have not been explored. High-energy astrophysical neutrinos travel over 100 Mpc, and they can explore new physics, which are extremely weakly coupled with neutrinos, such as LV. High-energy astrophysical neutrinos can also reach PeV energies and may enhance the new physics, in relation to power-counting nonrenormalizable operators [14].

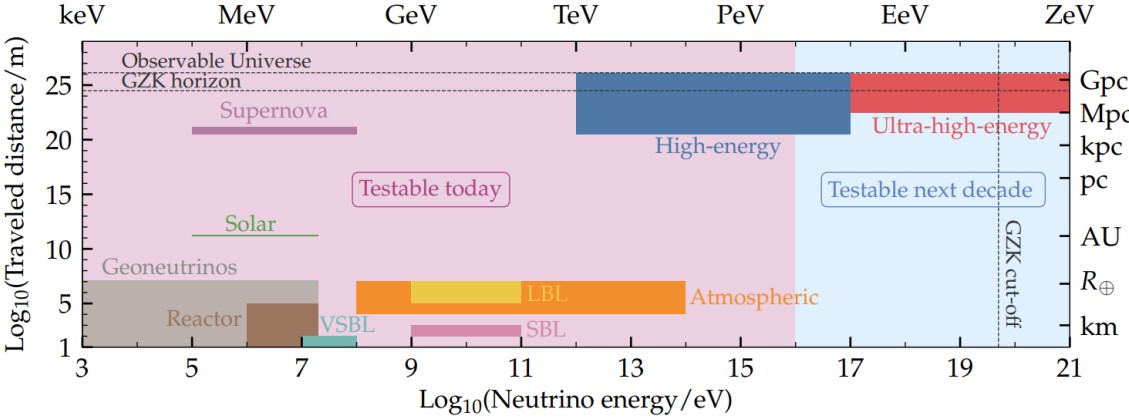

**Figure 1.** Neutrino sources are shown as a function of neutrino energy and distance traveled. Anthropologically-produced neutrinos, including reactor neutrinos, very-short-, short-, and long-baseline (VSBL, SBL, and LBL) neutrino beam experiments, can investigate new physics, in relation to short travel distance. Low-energy astrophysical neutrinos can be used to study new physics, related to longer travel distance. High-energy astrophysical neutrinos explore the highest energies and longest traveled distance, which is in the top right corner of this figure. Figure is adapted from Ref. [16].

## 2. Tests of Lorentz Violation with Kinematic Observables

High-energy particles, such as gamma rays [11,17–27] and UHECRs [12], have been used to test LV. Similarly, neutrinos can be used to find exotic effects, if they exist, with two advantages. First, neutrinos are elementary particles, while UHECRs are composite, in fact, with unknown composition. This makes the interpretation of LV constraints with neutrinos easier to interpret on a theoretical framework. Second, high-energy gamma rays interact with the cosmic microwave background and, thus, have shorter distances traveled than high-energy astrophysical neutrinos. The effects of LV in high-energy neutrinos arise from them experiencing the effect of a non-trivial vacuum as they propagate from their sources to us. A field in vacuum, motivated by new physics, could permeate space-time and violate the large-scale isotropy of the universe and, hence, produce LV (effectively similar to the classical æther). Under such conditions, neutrinos would emit particles in vacuum [28–32], and this energy loss would attenuate the highest energy neutrinos as they travel long distance. Such a test has been performed, using the high-energy astrophysical neutrino samples [33,34]. Multi-messenger astronomy allows us to study the difference of time-of-flight (ToF) between neutrinos and photons, beyond the neutrino mass effect. The first such opportunity was the supernova 1987A, where data was used the tests of Lorentz invariance [35–37]. These tests may be more interesting to use high-energy astrophysical neutrinos, due to energy dependencies of some models, and the first such opportunity was the blazar TXS0506 + 056, the first identified high-energy astrophysical neutrino source [33]. From the observation of the neutrino and photon arrival times, several limits of neutrino velocity deviation from the speed of light were derived [38–40].

Despite that TXS0506 + 056 has, so far, been the only detected source, searches for neutrino emissions from transient events are continuously performed. These analyses usually assume that neutrinos are travelling at the speed of light; however, if the neutrino ToF is modified due to LV, one could find more coincidence with transient events by assuming LV. The couplings of neutrinos and LV background fields can be classified into two groups, CPT-odd and -even types. The CPT-odd LV fields would change its sign in Lagrangian operators and, hence, effectively violate CPT transformation. If the effective velocity of neutrinos is larger than in the Lorentz-invariant case, that of antineutrinos is lower and vice versa. The IceCube collaboration has not identified any high-energy astrophysical neutrino point sources, except TXS0506 + 056, with high significance [41] (see [42] for a recent search of potential high-energy neutrino sources). In particular, no gamma ray burst (GRB) has been identified as a high-energy astrophysical neutrino source [43,44] under the standard model assumptions. However, by assuming energy dependent couplings with LV and sign changes (time delay or time advanced), it is possible to find coincidences with GRBs, with the scale of new physics order $\sim 10^{17}$ GeV [45] (See [46], regarding the implication of this results in the charged lepton sector). Although this is a very tantalizing result, it is challenging to verify it experimentally because the neutrino and anti-neutrino cross-section difference is less than 15% at $\geq 200$ TeV [47–49], and the charge separation is possible only in special reactions, such as resonant *W*-boson production [50]. In the near future, data from IceCube and neutrino observatories, currently under construction, such as KM3NeT [51] and GVD [52], will provide increased sensitivity to Lorentz violation. Further along, a generation neutrino telescopes on ice (IceCube-Gen2 [53]), water (P-ONE [54]), mountains (Ashra NTA [55], TAMBO [56], GRAND [57]), or outer space (POEMMA [58]), among others, will be able to test this hypothesis further.

### 3. Tests of Lorentz Violation with Neutrino Oscillations

Neutrinos are natural interferometers, which are able to measure extremely small quantities—such as the difference in neutrino masse—by observing the *beats* of the different neutrino flavors. Searches for distortions, arising from LV in the neutrino oscillation pattern, have been performed by almost all neutrino oscillation experiments [6]. Among them, natural sources, such as solar neutrinos, atmospheric neutrinos, and astrophysical neutrinos, have advantages, due to very-long-baseline and/or higher attainable energy. Atmospheric neutrinos can be produced at the other side of the Earth and penetrate the Earth's diameter (=12,742 km), and provide the largest possible interferometer on the Earth to search for neutrino oscillations. The energy of these neutrinos reaches order 50 TeV or more [59], which corresponds to the highest energy neutrinos produced by particles arriving at the Earth's surface. The atmospheric neutrino flux below around 50 TeV is produced predominantly from pion and kaon decays, and this is called the "conventional" atmospheric neutrino flux. This flux is relatively well understood and has being measured, compared with high-energy atmospheric neutrinos produced by charmed meson decay, which are predicted with larger errors and have avoided detection so far. Furthermore, the astrophysical neutrino flux starts to overtake atmospheric neutrino flux from around 50 TeV. Thus, these conventional atmospheric neutrinos can be used to search LV.

The concept of such a search is shown in the Figure 2, left. The LV oscillatory effect can be searched for in two ways, depending on the assumptions of the largest non-zero LV terms. On one hand, Super-Kamiokande [60] and IceCube-40 (partially instrumented IceCube) [61] searched for signatures of anisotropy in atmospheric neutrinos due to LV. On the other hand, AMANDA-II [62] and IceCube [63] looked for the spectral distortions due to LV. Figure 2, right, shows the effect of spectrum distortion, due to the presence of an interaction between neutrinos and an isotropic LV background field. One can see the very high sensitivity of this approach, especially for high-dimension LV operators. Here, atmospheric neutrinos detected by IceCube are sensitive to dimension-six LV operators, down to $\sim 10^{-36}$ GeV$^{-2}$, making these neutrinos provide one of the most sensitive probes for LV.

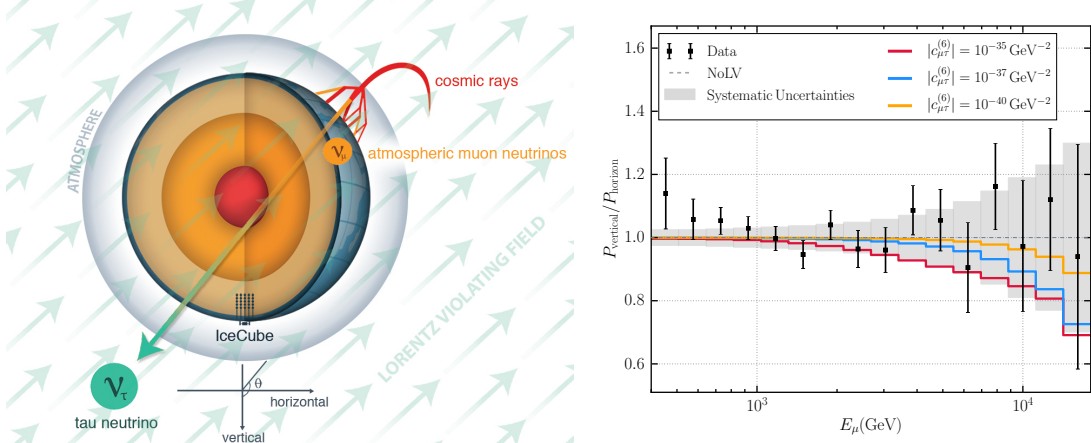

**Figure 2.** Left, artistic illustration of the search for LV with atmospheric neutrino oscillations. Atmospheric neutrinos are produced in the upper atmosphere, and their flavors may be converted due to couplings between neutrinos and LV background fields as they propagate. The effects induced by the new physics are negligible if neutrinos travel a short distance, namely for neutrinos entering the IceCube volume from the horizontal direction. However, neutrinos are produced near the northern sky travel a long distance before they reach IceCube, and they are significantly more impacted by interactions with the LV background fields. Right, expected atmospheric neutrino oscillation probability ratio with function of energy due to LV. Here, the vertical axis is the double ratio of oscillation probability for neutrinos, from the vertical to horizontal direction. No LV is adjusted to the unity in this figure. Nonzero LV modifies this ratio, and the larger deviations happen with the larger LV couplings. The figure shows the sensitivity to $|c_{\mu\tau}^{(6)}|$, one of dimension-six operators, which parameterize LV and this analysis is sensitive with. Figures are adapted from [63].

To use the highest energy atmospheric neutrino data, IceCube used the up-going data sample for this analysis [64]. These muons are created by neutrino interactions in the rock surrounding the detector or ice in the Antarctic glacier. The signal of LV is exhibited as a spectrum distortion in the high-energy muons. As you see from Figure 2, right, the data is consistent with unity, and there is no obvious sign of LV. This search does not find nonzero LV and produces a limit that reaches down to $\sim 10^{-24}$ GeV for the dimension-three operator or $\sim 10^{-36}$ GeV$^{-2}$ for the dimenion-six operator [63]. These are among the best constraints on LV from table-top experiments to cosmology.

## 4. Tests of Lorentz Violation with Neutrino Mixings

Astrophysical neutrinos are extremely long baseline neutrino oscillation experiments. In these systems, neutrino coherence depends on the details of the astrophysical neutrino source, detection method, and distance of propagation. For example, for the observed high-energy astrophysical neutrino flux, which is dominated by extragalactic sources, neutrino oscillations are not observable, due to the relative poor energy resolution and extremely large ratio of baseline to energy. In this scenario, we are only able to observe neutrino mixing, instead of oscillations, among neutrino states. However, even if we cannot resolve the neutrino oscillation pattern, new physics effects, such as LV, remain observable. This is because the propagation and detection eigenstates are not the same.

The SNO experiment searched for LV from the annual modulation of the solar neutrino signals [65]. By assuming a non-isotropic static LV background field within the solar system, neutrinos propagating one direction may be affected differently than others. The search of such a signal must control all other natural modulations of solar neutrinos, including the eccentricity of the Earth's orbit and day–night caused by the Earth matter propagation. The sensitivity of this search reaches order $\sim 10^{-21}$ GeV, in dimension-three LV coefficients.

High-energy astrophysical neutrinos offer even longer baselines. Most of these neutrinos do not have identified sources, and the flux is isotropic. Additionally, the source candidates populate distances from the Earth on the order 100 Mpc or more. In the high-energy starting event (HESE) sample, by IceCube [34], the energy of these neutrinos ranges

from around 60 TeV to 2 PeV. These high-energy neutrinos can push the search of higher-dimension LV operators. Figure 3, left, shows the sensitivities of different systems to LV operators. High-energy astrophysical neutrino flavors are expected to offer the most sensitive LV searches in dimension-five and dimension-six operators [66].

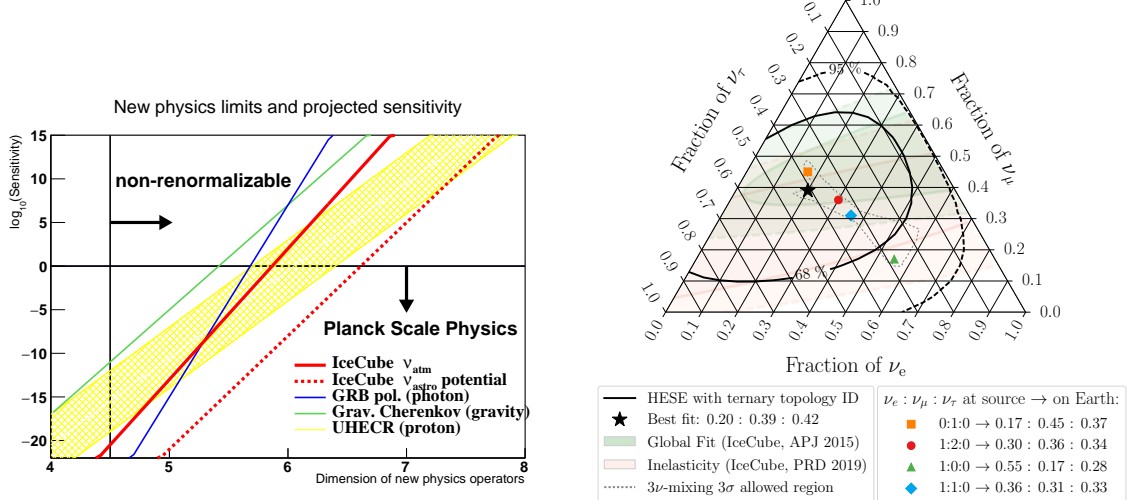

**Figure 3.** Left, the LV sensitivities of different systems. The sensitivity is normalized with the Planck energy ($E_P = 1.22 \times 10^{19}$ GeV) assuming LV arises from Planck-scale physics. This means that the natural scale of a dimension-five LV operator is $1/E_P$, a dimension-six LV operator is $1/E_P^2$, and so on, and the sensitivity is normalized, so that these numbers appear to have unity. Lines are gravitational Cherenkov emission [67,68], GRB vacuum birefringence [69], UHECRs [12], atmospheric neutrino oscillations [63], and the expected sensitivity from the high-energy astrophysical neutrino flavors. Figure is adapted from [66]. Right, HESE flavor ratio ($\nu_e : \nu_\mu : \nu_\tau$) measurement by IceCube. Each corner represents electron, muon, and tau neutrino dominant state. The standard scenarios (dotted line area) give roughly ($\nu_e : \nu_\mu : \nu_\tau$)~(1:1:1) on the Earth, regardless of the assumption of the flavor ratio at the source. This means the expected flavor ratio on the Earth is always around the center of this plot, in standard assumptions. On the other hand, current data contours enclose a large area, so different standard scenarios cannot be distinguished. Figure is adapted from [70].

Figure 3, right, shows the current status of the high-energy neutrino flavor ratio measurements. Since the statistics of high-energy neutrinos are low, flavors are measured integrated over the whole spectrum and normalized with the total flux, namely the flavor ratio, ($\nu_e : \nu_\mu : \nu_\tau$), is reported. The astrophysical neutrino production models give some information about the neutrino flavor composition at the source. The most likely scenario is that there is some combination of electrons and muon neutrinos. Two extreme cases are productions of astrophysical neutrinos, which are dominated by electron (1:0:0) or muon neutrinos (0:1:0). All other possibilities are between these two models. Remarkably, all of these scenarios make more or less the same flavor ratio at the Earth by neutrino mixing, namely ~(1:1:1) [71,72]. All of these predict the flavor ratio at the Earth is near the center of the flavor triangle and the spread of the center region is related to the current uncertainty of the mixing angles; for projections see [73]. On the other hand, current data encloses a large region and it is not easy to distinguish any particular scenario [70,74–76]. Thus, it is necessary to shrink this contour to measure possible deviation of the flavor ratio from the standard case. This requires larger sample sizes and better algorithms to measure neutrino flavors in neutrino telescopes [73]. Many different types of new physics can be discovered from the effective operator approach with astrophysical neutrino flavor measurement, including neutrino-dark matter interactions [77–79], neutrino-dark energy interactions [80,81], neutrino self-interactions [82–84], and neutrino long-range forces [85]. The first IceCube results to test these models were published recently [86].

To summarize, the search for signatures of LV, with high-energy astrophysical neutrinos, has just begun. The sensitivity to certain operators seems to exceed any known

systems and reaches the Planck scale. Thus, the study of these probes has a great potential for the discovery of violations of fundamental space-time symmetries.

**Funding:** CAA is supported by the Faculty of Arts and Sciences of Harvard University and the Alfred P. Sloan Foundation. TK is supported by the Science and Technology Facilities Council (UK).

**Data Availability Statement:** Data are available in corresponding references.

**Acknowledgments:** We thank to Rogan Clark for the careful reading of this manuscript. CAA is supported by the Faculty of Arts and Sciences of Harvard University and Alfred P. Sloan Foundation. TK is supported by the Science and Technology Facilities Council (UK).

**Conflicts of Interest:** The authors declare no conflict of interest.

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
