# Peer review of "Lorentz Symmetry and High-Energy Neutrino Astronomy"

_universe, doi:10.3390/universe7120490_

Round 1

Reviewer 1 Report

  1. This short review motivates experimental searches for Lorentz invariance violations (LIV) using high-energy neutrino astronomy, including searches from astrophysical and atmospheric neutrinos, and comparing the relative sensitivities. One aspect that was not discussed,  although in principle can constitute a very sensitive probe of LIV, is the core collapsed supernovae.  I leave it optional to the authors to mention this test as well, which was discussed in  J.~Ellis, H.~T.~Janka, N.~E.~Mavromatos, A.~S.~Sakharov and E.~K.~G.~Sarkisyan, %``Probing Lorentz Violation in Neutrino Propagation from a Core-Collapse Supernova,'' Phys. Rev. D \textbf{85}, 045032 (2012) doi:10.1103/PhysRevD.85.045032 [arXiv:1110.4848 [hep-ph]].

    and compare the potential sensitivity of this probe with
    the other probes mentioned in the brief review.

    The review is certainly well written, comprehensive and should be published in
    this special issue.

Author Response

This short review motivates experimental searches for Lorentz invariance violations (LIV) using high-energy neutrino astronomy, including searches from astrophysical and atmospheric neutrinos, and comparing the relative sensitivities. One aspect that was not discussed,  although in principle can constitute a very sensitive probe of LIV, is the core collapsed supernovae.  I leave it optional to the authors to mention this test as well, which was discussed in  J.~Ellis, H.~T.~Janka, N.~E.~Mavromatos, A.~S.~Sakharov and E.~K.~G.~Sarkisyan, %``Probing Lorentz Violation in Neutrino Propagation from a Core-Collapse Supernova,'' Phys. Rev. D \textbf{85}, 045032 (2012) doi:10.1103/PhysRevD.85.045032 [arXiv:1110.4848 [hep-ph]].

and compare the potential sensitivity of this probe with

the other probes mentioned in the brief review.

The review is certainly well written, comprehensive and should be published in

this special issue.

 *Thank you for your comment, we added a short paragraph to discuss supernova neutrinos including the recommended reference.

Reviewer 2 Report

The manuscript “Lorentz symmetry and high-energy neutrino astronomy” reviews the current status of searches for Lorentz violation with a focus on recent results and prospects of using high-energy neutrinos to search for violation of Lorentz symmetry. The article is very well written and summarizes relevant measurements. The complexity involved in understanding limits on new physics operators or how different experimental bounds relate to each other is only touched upon in the article and could not be reviewed given the length of the article. Hence, the review provides a high-level summary, without going into details about effective operators, which is left to the reader to explore following the references given. 

This review is worth publication and very timely. I have only a few minor comments:

The review should include studies of LV based on SN1987A, while these are not done with high-energy neutrinos, they still remain of interest to the reader and provide the historic background of using astrophysical neutrinos on long baselines. For example: Longo, M. J. 1988, Physical Review Letters, 60, 173, Krauss, L. M., & Tremaine, S. 1988, Physical Review Letters, 60, 176

Abbreviations such as GZK, VSBL in Fig 1 or notations such as c_mu_tau^{(6)} might not be clear to readers without background knowledge or going to the sources of the figures. If they can be intelligently be included in the text, it will make the manuscript easier understandable for the common reader.

The list of upcoming neutrino observatories at the top of page 3 lists experiments that at a very different level of maturity and funding. TAMBO is in the early concept stage and does not fit in the list. If the authors wish to list Earth skimming neutrinos experiments, such as TAMBO, then those should be listed separately, but other such experiments should be included.

top of page 4:
“the IceCube” —> “IceCube”

Author Response

Dear Reviewer,

Thank you for your suggestions for this paper. Authors' reply to you is in the attachment.

Kind regards,

Reviewer 3 Report

Overall, a nice, brief review (informative yet concise) of astrophysical neutrinos for Lorentz invariance violation searches.

I understand that this is a brief review of the particular field, as there is no original work presented. If this is the case, I recommend acceptance after minor revisions.

I have the following comments and suggestions to the authors:

General suggestions: 
Clean up the language (experiments…systems repeated in section 1, section 4 also has some repetitive formulations on the expected flavor ratio), run a spellcheck (IceCube or the IceCube neutrino observatory, not the IceCube, etc).

The introduction could benefit from more details on how LV can affect neutrinos, and on the connection between the higher-dimensional operators, LV and the theories predicting them. And also on astrophysical neutrino production and propagation. Especially since this is meant for a broader audience who might not be familiar with LV or astrophysical neutrinos.

The authors should consider discussing TDEs as possible ways to search for LV with astrophysical neutrinos, and why or why not this might be promising. I don't understand why GRBs are singled out, but TDEs are never mentioned.

The future outlook is extremely short. A projection of achievable limits in the next decade would be helpful.

Detailed remarks:
Section 2. "
Such a test has been performed using the diffused high-energy astrophysical neutrino sample [19]." This statement is misleading: [19] is a special selection of astrophysical neutrinos, but not "the diffuse" sample. Further, in [19] no tests have been performed on LV. Also, "HESE" should be introduced here (instead of in section 4).

Section 2. "The IceCube collaboration has not identified any high- energy astrophysical neutrino point sources except TXS0506+056 [19]." Again, a misleading statement: [19] is just one search, and not the most sensitive one. The analysis in Phys. Rev. Lett. 124, 051103 (2020), finds 2.9σ significance for  NGC 1068 (post-trial). Further, TDEs have been identified as likely sources of high-energy neutrinos, see Nature Astronomy volume 5, pages510–518 (2021). 

Section 2. Please elaborate at what energy the neutrino and anti-neutrino cross-sections become ~identical, to make it easier for the reader to put the atmospheric fluxes further down in perspective.

Section 3. In addition to poorly understood charm flux, also astrophysical neutrinos become a significant part of the flux above 50-ish TeV.

Section 3. "To use the highest energy atmospheric neutrino data, the IceCube used the up-going data sample for this analysis [41]." The authors should describe this sample in at least the bare necessary detail, namely that it consists of muons entering the detector, which stem from muon neutrinos that interacted on their way through the Earth not too far from the detector.

Section 4 / Fig. 3, right: The standard scenarios are marked in the figure by the dotted line.

Author Response

reply attached

Reviewer 4 Report

Please see the PDF attached.

Author Response

reply  attached
